# Gold Nanoparticle Virus-like Particles Presenting SARS-CoV-2 Spike Protein: Synthesis, Biophysical Properties and Immunogenicity in BALB/c Mice

**DOI:** 10.3390/vaccines12080829

**Published:** 2024-07-23

**Authors:** Vivian A. Salazar, Joan Comenge, Rosa Suárez-López, Judith A. Burger, Rogier W. Sanders, Neus G. Bastús, Carlos Jaime, Joan Joseph-Munne, Victor Puntes

**Affiliations:** 1Vall d’Hebron Institut de Recerca, 08035 Barcelona, Spain; vivianangelica.salazar@icn2.cat (V.A.S.); joan.comenge.ext@vhir.org (J.C.); 2Networking Research Centre for Bioengineering, Biomaterials and Nanomedicine (CIBER-BBN), Instituto de Salud Carlos III, 28029 Madrid, Spain; neus.bastus@icn2.cat; 3Departament de Química, Universitat Autònoma de Barcelona, Bellaterra, 08193 Barcelona, Spain; rosa.suarez@uab.cat (R.S.-L.); carlos.jaime@uab.cat (C.J.); 4Department of Medical Microbiology and Infection Prevention, Amsterdam University Medical Centers, Location AMC, University of Amsterdam, Amsterdam Infection & Immunity Institute, 1105 AZ Amsterdam, The Netherlands; j.a.burger@amsterdamumc.nl (J.A.B.); r.w.sanders@amsterdamumc.nl (R.W.S.); 5Institut Català de Nanociència i Nanotecnologia (ICN2), CSIC and BIST, Campus Universitat Autònoma de Barcelona, 08193 Barcelona, Spain; 6Department of Microbiology, Hospital Universitari Vall d’Hebron, 08035 Barcelona, Spain; 7Institució Catalana de Recerca i Estudis Avançats (ICREA), 08010 Barcelona, Spain

**Keywords:** virus-like particles (VLPs), gold nanoparticles, spike protein SARS-CoV-2, SARS-CoV-2 vaccine

## Abstract

Gold nanoparticles (AuNPs) decorated with antigens have recently emerged as promising tools for vaccine development due to their innate ability to provide stability to antigens and modulate immune responses. In this study, we have engineered deactivated virus-like particles (VLPs) by precisely functionalizing gold cores with coronas comprising the full SARS-CoV-2 spike protein (S). Using BALB/c mice as a model, we investigated the immunogenicity of these S-AuNPs-VLPs. Our results demonstrate that S-AuNPs-VLPs consistently enhanced antigen-specific antibody responses compared to the S protein free in solution. This enhancement included higher binding antibody titers, higher neutralizing capacity of antibodies, and stronger T-cell responses. Compared to the mRNA COVID-19 vaccine, where the S protein is synthesized in situ, S-AuNPs-VLPs induced comparable binding and neutralizing antibody responses, but substantially superior T-cell responses. In conclusion, our study highlights the potential of conjugated AuNPs as an effective antigen-delivery system for protein-based vaccines targeting a broad spectrum of infectious diseases and other emergent viruses.

## 1. Introduction

Infectious diseases, including those caused by multidrug-resistant organisms and zoonosis, remain a major threat to global health and health security. This challenge has stimulated the exploration of new vaccine strategies and antigen-presenting platforms, aiming not only at enhancing efficacy, but also ensuring accessibility, affordability, and rapid development as soon as a new threat is detected. Vaccine development has evolved significantly from antigen and adjuvant materials simply mixed and co-administered as a bolus injection to actual mRNA vaccines. First-generation vaccines have proven to be suboptimal and not effective enough to protect the population against some pathogens. In addition, many pathogens with pandemic potential exhibit variability in their surface antigen composition, and novel technologies are required to develop safe vaccines against each new variant within a short timeframe [1] while providing long-term robust humoral and cellular responses [2]. For example, although the available vaccines have demonstrated high efficacy in mitigating the COVID-19 pandemic, there are still some challenges remaining, from optimal storage conditions to achieving a long-lasting immune response.

It has been proposed that next-generation vaccines should consider overall vaccine structure in addition to composition, assuming that the overall supramolecular structure is as crucial as the training elements it contains [3]. This is because the key feature of immune cells that enables them to detect and categorize infection is their repertoire of pattern-recognition receptors, complement molecules, and antibodies, which bind certain general types of molecules that are expressed across broad classes of pathogens but are absent in our healthy cells. However, this can be slightly misleading, as some of these molecules are not actually restricted in their expression to pathogens. Instead, what distinguishes a pathogen as non-self is the structural conformation of its molecules or their subcellular location. Following these principles, it is widely understood that the artificial display of peptides, proteins, or antigens in an ordered, repetitive manner (epitope repetition) induces an enhanced immune response compared to the “free” protein antigen [4].

In this sense, the integration of virus structural characteristics can be achieved through the rational design of virus-like particles (VLPs) where structure becomes the adjuvant. VLPs are constructs composed of a nanometric core exposing repetitive and densely packed viral or bacterial antigens [5,6]. This design facilitates the activation of a robust immune response [7] where they exhibit the ability to induce both humoral and cellular immune responses [8]. Initially, self-assembling viral proteins into particulate structures resembling intact viruses was described by Caspar and Klug (1962). This has garnered increasing recognition as a promising candidate for vaccine development since then, including the human papillomavirus vaccine. With the advent of nanotechnology in the early 2000s, gold nanoparticles (AuNPs) have emerged as a particularly promising antigen-presenting platform thanks to their biocompatibility [9] optical properties, and tunable size and surface properties [10,11]. This has paved the way for engineering VLP-based therapeutics with augmented physicochemical properties and multifunctionality [12]. The interaction between proteins and inorganic surfaces, particularly gold nanoparticles, has been extensively investigated under the term Protein Corona [13,14,15,16,17]. This phenomenon involves the formation of a single layer of densely absorbed proteins that provides a unique biological identity to the NP [18]. 

Accordingly, in addition to antigen presentation, research has shown that the NP structure plays a pivotal role in determining biodistribution, persistence, and ultimately efficacy [3]. Vaccine biodistribution, mainly governed by the NP physicochemical properties, significantly influences the type of immunological response generated. As naïve T and B cells recirculate through the lymph nodes and spleen, vaccines that can access them will confer a more robust humoral and cell immunity [19]. It has been shown that large NPs (>100 nm) remained in the injection site after intradermal injection, whilst their smaller counterparts migrated to the lymph nodes, promoting a more robust complete immunity [20,21]. Consistently, a wide range of NPs have been used as carriers for antigens and adjuvants [22], and the effects of NP’s size, shape, and surface charge on biodistribution and cellular uptake investigated [23,24]. These studies have underscored the critical importance of precisely controlling the spatiotemporal codelivery of antigens and adjuvants to successfully mobilize the immune system against a desired target [25]. It indicates that the nanoscale structure, chemistry, and composition of immunostimulatory compounds can be modulated to maximize cellular uptake, antigen processing, and antigen-presenting cells. These processes are partially governed by particle physicochemical characteristics and eventually determine the immune response. Furthermore, the density of antigen on VLPs also influences the type of immune response, with a dramatic effect in T helper cell-dependent responses [26]. Hence, the total number of antigens per particle, as well as their spacing and arrangement, critically influence the immunologic response conferred by VLPs [19]. 

In this work, we addressed these crucial needs by preparing and characterizing SARS-CoV-2 spike protein (S) conjugated to 50 nm AuNPs as a model example. We also have evaluated in vivo the specific SARS-CoV-2 immunogenicity after immunization of BALB/c mice with S-AuNPs-VLPs. In addition, we have measured in vitro virus neutralization, which is extremely well correlated with in vivo protection in humans [27]. Several studies have explored the conjugation of derived coronavirus proteins with AuNPs, highlighting their potential in mitigating in vivo viral infections. Chen et al. demonstrated a straightforward method for coating viral antigens onto Au NPs, showing promise for enhanced vaccine development [28]. Similarly, Sekimukai et al. observed a strong antigen-specific IgG response against severe acute respiratory syndrome-related coronavirus infections with similar NPs but failed to induce protective antibodies and limit eosinophilic infiltration in lungs in infected mice [29]. In our system, we selected NP–viral protein incubation conditions that promote cooperative absorption to obtain dense ordered patterns (mosaic-like) of viral protein on the NP surface. This is achieved by controlling the key parameters affecting NP–protein interactions, such as relative concentrations, incubation media, pH, and temperature [30]. Physicochemical assays and molecular dynamic simulation confirmed the cooperative adsorption of proteins on the AuNP surface, preserving antigen structure, and maintaining colloidal stability under physiological conditions, essential for ensuring an optimal immune response [31]. Subsequently, the immunogenicity of S-AuNP-VLPs was assessed in BALB/c mouse models. We demonstrated that AuNPs-VLPs presenting the Wuhan strain (D614G) spike protein could induce SARS-CoV-2-specific binding and neutralizing antibodies and specific T-cell responses. This study provides chemical tools for the rational design of a simple and straightforward strategy that may be worth supporting in the global effort to develop the next-generation vaccine against SARS-CoV-2 and other emergent viruses.

## 2. Materials and Methods

### 2.1. Synthesis of AuNPs

A solution containing 2.2 mM sodium citrate (SC) in 150 mL of Milli-Q water was heated using a heating mantle in a 250 mL three-necked round-bottomed flask for 15 min with vigorous stirring. To prevent solvent evaporation, a condenser was employed. Five minutes after boiling started, 1 mL of HAuCl_4_ (25 mM) was injected. The color of the solution changed from yellow to bluish-grey, then to soft pink within 5 min. Immediately after synthesizing the Au seeds in the same reaction vessel, the reaction was cooled until the solution temperature reached 90 °C. Subsequently, 1 mL of HAuCl_4_ solution (25 mM) was injected. After 10 min, the reaction was completed. The 1 mL of HAuCl_4_ addition was repeated twice. Following this, the sample was diluted by extracting 55 mL of the sample and adding 53 mL of Milli-Q water and 2 mL of 60 mM sodium citrate. This resulting solution served as the seed solution, and the process was repeated until the particles reached the desired size.

### 2.2. Synthesis of S-AuNPs-VLPs

Full SARS-CoV-2 spike protein (Abyntek Cat: 40589-V08H4) was washed 3 times with water using an EMD Amicon™ 3 K 6000 g for 7 min at 4 °C to avoid undesired interactions of AuNPs with excipients. Finally, SARS-CoV-2 spike protein was dispersed in a borate buffer (10 mM, pH 9.0) at a concentration of 7.8 μg/mL. This solution was freshly prepared for each experiment. For AuNP functionalization, equal volumes of 7.8 mg spike/L in borate buffer and 50 nm AuNPs (1.99 × 10^11^ NP/mL) were mixed by slowly adding AuNPs to the protein solution under stirring. The solution was left for 72 h at 4 °C. The resulting AuNPs were dialyzed against 1 mM borate (pH 9.0) and readily concentrated to 5.6 × 10^12^ NP/mL with rotary evaporation at 30 °C and 100 rpm (IKA™ RV 8 V-C Rotary Evaporator).

### 2.3. Synthesis of Murine Albumin VLPs

The conjugation of the 50 nm AuNPs with murine albumin followed a process similar to that for SARS-CoV-2. Murine albumin was dissolved in borate buffer (10 mM, pH 9.0) at a concentration of 13.7 μg/L. Subsequently, AuNPs (1.99 × 10^11^ NP/mL) were added to the protein solution. The formed VLPs were then concentrated by centrifugation using an Amicon Ultra 10,000 KD filter.

### 2.4. UV-Vis Characterization 

UV-visible absorbance spectra were acquired on an Agilent Technologies Cary 60 UV-Vis Spectrophotometer. A 1/200 dilution of the different AuNP solutions was placed in a plastic cell with a 10 mm optical path length. The spectral analysis was performed between 300 nm and 800 nm at room temperature. Cary Win UV software by Agilent was used to collect the spectrophotometer output. Data were represented using Origin 8.0.

### 2.5. Dynamic Light Scattering (DLS) and Zeta Potential Analysis 

DLS and Zeta potential measurements were measured by dynamic light scattering (DLS) and zeta potential on a Malvern Zetasizer Nano ZS90, which incorporates a Z-potential analyzer (Malvern Instruments Ltd., Worcestershire, UK). A 1/100 dilution of AuNP was placed into a specific cuvette, and the measures were performed in triplicate. Malvern Zetasizer software was used to collect the instrument output. Data were represented using Origin 8.0.

### 2.6. Sodium Cyanide Assay 

The degradation of the AuNP gold core by sodium cyanide was assessed to confirm the functionalization of AuNP. Naked AuNP and conjugated AuNP (at conjugation concentration) were exposed to 15 mM sodium cyanide. UV-Vis spectral analysis was performed every minute, as explained above. The decay in absorbance at 532 nm was monitored over 30 min to assess the degradation state of the different AuNPs. 

### 2.7. NaCl Stability 

The stability of the functionalized AuNPs was confirmed by the addition of increasing concentrations of NaCl from 10 mM to 200 mM. The UV-Vis spectra of the naked and conjugated samples (1:100) were analyzed immediately after salt addition. All spectra were normalized to λ400 nm for better comparison using SpectraGryph 1.2—spectroscopy software.

### 2.8. Molecular Dynamics

**Building model from structural data.** The 3D structure of the S protein was obtained from the SWISS-MODEL database [32], with the model identified by the SMTL ID: 7cn8.1 [33]. To elucidate the overall charge distribution across the S protein, we utilized the UCSF Chimera software [34], as depicted in Appendix A. The AA protein model was subsequently coarse-grained to generate the CG model.

**(CG) model and Soft Repulsive Parameters (SRP) definition.** S protein was divided into 10 CG beads using the Shape-Based Coarse Graining [35] approach, a method within VMD [36] where large-scale motions of organic molecules are represented using as few CG sites as possible. The beads were color-coded to reflect their binding strengths to the AuNP, as suggested by the charge distribution on the original amino acid sequence. Dark red beads (H) denote the highest binding affinity, followed by red beads (M), and blue beads (L) representing the least attractive interaction. The AuNP itself was formed by the aggregation of gold beads (Au), and the surrounding solvent was depicted by water (W) beads. 

The effective interaction between CG beads in Dissipative Particle Dynamics (DPD) is defined by the Soft Repulsive Parameters (SRP) (Table 1). W/W was maintained at 25 in consistency with Groot and Warren [37]. To capture the hydrophobicity and compactivity of the AuNP, W/Au and Au/Au were set to 120 and 1, respectively. Finally, to simulate the behavior of S proteins in solution, we tested different SRPs between water and S protein (W/S protein) and between S proteins themselves (S protein/S protein). This testing resulted in values of 40 and 55, respectively, being identified as the most representative experimental behavior.

**Simulation condition.** The cooperative adsorption of S proteins onto the AuNP surface was studied by the DPD—Monte Carlo hybrid model. All simulations were performed in an NPT ensemble using a dimensionless unit system. Corresponding to the experimental setting, systems were solvated in a cubic box of 35 × 35 × 35 $d_{0}^{3}$ with about 90,000 beads of water, 25,000 beads of Au (~10 nm), and 150 S protein with Periodic Boundary Conditions (PBC) invoked. The time step used to integrate the equations of motion was Δ*t* = 0.03. The temperature was set to 0.42, as established in the simulations of lipid membranes. According to Groot and Rabone [38], it was assumed that the reduced mass and diameter of one bead is 1, and the cut-off distance $d_{0}=0.646\;nm$. The primary dimension of our S protein model, roughly 5 times smaller than the actual S protein size, measured approximately 3 nm. Consequently, the diameter of our AuNP was approximately 10 nm, maintaining a consistent size relationship with the model [39].

### 2.9. Bioconjugate Analysis by STEM-SEM

Scanning transmission electron microscopy (STEM) was utilized to visualize both naked AuNPs and conjugated AuNPs. The FEI MAGELLAN 400 L XHR SEM microscope enabled image acquisition in high-energy (15–30 kV) SEM and STEM modes. Ultrathin Formvar-coated 200-mesh copper grids (Ted-pella, Inc., Redding, CA, USA) were immersed directly in the sample solution and allowed to air dry overnight.

For the analysis of surface contact between AuNPs in the case of partial antibody surface passivation, TEM and SEM images of the S-AuNP-VLPs complexes were utilized. The charging effect observed on the edge of the image visualized by SEM is associated with the presence of the organic layer, providing information on the degree of AuNP surface passivation by the viral proteins. 

### 2.10. Immunization of Mice, Collection of Sera, and Isolation of Splenocytes 

This is a preliminary proof-of-concept study to demonstrate the immunogenicity of S-AuNPs-VLPs. The dose, administration route, and prime-boost interval of S-AuNPs-VLPs referred to previous studies that immunized mice with HPV:HIV VLPs for inducing antibody responses [40,41]. S-AuNPs-VLPs were emulsified with an equal volume of aluminum hydroxyphosphate sulfate (Thermo Fisher Scientific). All mouse groups had equal gender distribution (male *n* = 4 and female *n* = 4 per group) and were 8 weeks old. BALB/c mice were injected intramuscularly in the thigh with 100 µL of the different formulations: Group A: SARS-CoV-2 AuNPs-VLPs (5.6 × 10^12^ NP/mL, 10 µg protein); group B: free SARS-CoV-2 spike protein (10 µg protein); group C: Pfizer-BioNTech vaccine (100 ng mRNA, Cominarty injection. Original/Omicron BA. 4-5-Positive control); group D: borate buffer (naïve mice); group E: murine albumin AuNPs-VLPs (5.6 × 10^12^ NP/mL, 10 µg protein). Note that all formulations except Pfizer-BioNTech vaccine were previously mixed 1:1 with Alum adjuvant before injection. The prime-boost interval was 2 weeks. Mice were sacrificed on day 28. Blood samples were collected from the heart of mice. Sera were recovered by centrifugation and stored at −20 °C for ELISA and neutralization assay. Murine spleens were removed and pressed individually through a cell strainer (Falcon) with a 5 mL syringe rubber plunger. Following the removal of red blood cells with RBC Lysis Buffer (Thermo Fisher Scientific), splenocytes were washed and resuspended in RPMI 1640 supplemented with 10% fetal bovine serum (FBS), penicillin-streptomycin, and 20 mM HEPES.

### 2.11. Enzyme-Linked Immunosorbent Assay

To assess the specificity of the IgG produced by the immunized mice, recombinant whole SARS-CoV-2 protein was used for enzyme-linked immunosorbent assays. Briefly, blood samples were collected at 28 days after prime and boost intramuscular administration of the antigen as described previously. Serum samples (100 uL dilution 1:1000) were incubated for 1 h at room temperature. After three washes, 100 μL of rabbit anti-mouse IgG:HRP conjugate was added to each well and kept for 1 h. Following three washes, 100 μL TMB substrate solution was added to each well and incubated in the dark at room temperature for 10 min. The reaction was stopped by adding 100 μL of TMB stop solution. The OD was measured at 450 nm by using a Varioskan absorbance microplate reader, according to the manufacturer protocol. 

### 2.12. Determination of Neutralizing Antibodies

To evaluate in vivo the specific SARS-CoV2 immunogenicity after immunization of BALB/c mice with S-AuNPs-VLPs, we quantified the presence of neutralizing antibodies. Briefly, SARS-CoV-2 virus neutralization assays were performed as described previously [42,43]. Duplicates of two-fold serial dilutions (starting at 1:10) of heat-inactivated sera (30 m, 56 °C) were incubated with 100 median tissue culture infectious dose of SARS-CoV-2 strain D614G (WT) at 35 °C for 1 h in 96-well plates. Vero E6 cells were added in a concentration of 20,000 cells per well and incubated for 72 h at 35 °C. The serum virus neutralization titer was defined as the reciprocal value of the sample dilution that showed a 50% protection of virus growth.

### 2.13. Antigen-Specific IFN Gamma Production Analysis 

On day 28 post-immunization, mouse spleens were minced and passed through a 70 μm cell strainer (Corning) to obtain single-cell suspensions. Red blood cells (RBCs) were lysed using an RBC lysis buffer (Invitrogen 00-4333-57), and cells were resuspended in RPMI 1640 medium containing 10% FBS. Viable cells were determined by trypan blue staining. Then, 2 × 10^6^ splenocytes were plated in a 24-well plate and were stimulated with 1.4 μg of purified whole SARS-CoV-2 proteins for 72 h at 37 °C. Supernatants were collected and stored at −20 °C until INF-γ quantification. The concentration of IFN-γ in each supernatant was quantified by duplicates according to manufacture guidelines (Abyntek, E1122 kit). 

### 2.14. Statistical Analysis 

Statistical analysis and graphing were carried out in GraphPad Prism 9.4.0 software. Statistical differences were analyzed using two-way ANOVA adjusted for multiple comparisons

### 2.15. Ethics Statements

Six- to eight-week-old BALB/c mice were purchased from Envigo (an Inotiv company, Chicago, IL, USA) and approved by local authorities (Generalitat de Catalunya, project number 11157) and Universitat Autònoma de Barcelona Ethics Commitee. The animal experiments strictly conformed to the animal welfare legislation of the Generalitat de Catalunya. All the experiments were approved by the local Research Ethics Committee (Procedure 43.19, Hospital de la Vall d’Hebron, Universitat Autònoma de Barcelona).

## 3. Results

### 3.1. Synthesis and Biophysical Properties of S-AuNPs-VLPs

Our goal was to develop mosaics of SARS-CoV-2 spike protein from variant D164G (Wuhan) on top of 50 nm gold nanoparticles (AuNPs) for their presentation to the immune system. The full spike protein (S) was selected since it is involved in cell entry, and thus a logical vaccine target for attempting sterilizing immunity. Remarkably, it presents a particular challenge; it has been reported that differently to the vast majority of RNA viruses, the separation between spike proteins in SARS-CoV-2 is around 25 nm, far from the 5–10 nm needed for enhanced immune activation, since epitope repetition boosts immunogenicity [4,5,6], which might explain the inefficient and short-lived neutralizing antibody responses after vaccination and infection with SARS-CoV-2 [44], highlighting the importance of structure in antigen presentation. Under these circumstances, presenting a dense mosaic of S protein to train our immune system is particularly appealing. The size of AuNPs was chosen to mimic the size of SARS-CoV-2 and to accommodate a substantial number of proteins on their surfaces. 

Citrate-stabilized AuNPs were synthesized via a seeded-growth multistep method based on the aqueous reduction of Au salts by sodium citrate (SC) [45]. Once synthesized, AuNPs were purified and dispersed in a 2.2 mM SC solution. The highly monodisperse AuNPs of 50 ± 6 nm in diameter (Figure 1A) were obtained at a concentration of 5 × 10^10^ NPs/mL. These AuNPs showed a distinct surface plasmon resonance (SPR) absorption peak at 529 nm (Figure 1B) and exhibited a surface charge of −44 mV (Figure 1C). These AuNPs were subsequently added dropwise to the lyophilized viral protein dissolved in borate buffer 10 mM for 48 h under gentle stirring at 4 °C. We adapted conjugation conditions previously described by our group to promote cooperative absorption of proteins at the NP surface [30]. Considering the available surface per NP (7854 nm^2^ per 50 nm AuNPs) and estimating a spherical projection of the spike protein of 36.31 nm^2^, as many as 173 proteins per AuNP could be loaded. After this estimation, we modeled Langmuir-type isothermal adsorption curves to identify conditions and concentrations needed to obtain surface-saturated S-AuNP conjugates (Figure 1D). Thus, AuNPs were exposed to varying amounts of protein, after which the supernatant was analyzed, and the protein concentration in the supernatant plotted against the shift in the SPR band indicative of protein absorption onto the AuNP [14]. Upon the full replacement of citrate molecules by proteins, the SPR peak red-shifted until reaching a plateau. This redshift of the SPR band is directly correlated with the increased compactness of the dielectric layer surrounding the AuNP surface and continues to increase as long as there is space for the incorporation of more proteins at the NP’s surface. Once the NP surface is saturated, the SPR peak remains unaltered even if protein concentrations are further increased. Of note, adsorption of S protein onto the gold surface is a non-static process in which the system evolves towards an equilibrium, which is influenced at any moment by both the availability of gold surface and the interactions with neighboring S proteins. Given the appropriate conditions described in the manuscript, the system reaches a final state in which clustering of proteins onto the gold surface stabilizes the proteins at the NP surface in a well-described phenomenon named hardening of the protein corona, where initially loosely bound proteins evolve to a permanent coating of proteins thanks to coordination bonds with the NP surface and crowding effects between proteins [18]. Note that according to Langmuir-type isothermal adsorption curves, the ratio 1:200 was chosen to ensure the high density of proteins needed to guarantee steric hindrance, preventing aggregation and non-specific interactions, whilst cooperative effects are promoted. According to the isothermal adsorption, ~400 proteins in solution per AuNP are needed to reach saturation of the AuNP surface (an excess of protein is always needed to displace the equilibrium towards NP surface saturation [14]). We hypothesized then that a densely coated but un-saturated NP surface, made of dense protein domains, would make antigens more available (reactive) for processing, while preventing NP aggregation (for coating rates above 60%) [18], would be a neat VLP candidate. This led to a chosen incubation condition of 200 proteins/AuNPs. 

Upon mixing, the system slowly evolved towards a new equilibrium state where NPs were coated by the viral proteins. The incubation of NPs with proteins is challenging [18]; normally, the low ionic strength where NPs are dispersed induces protein denaturalization, while high salty media where proteins are soluble induce NP irreversible aggregation and sedimentation. To promote their cooperative adsorption on the surface of the NP, correct conjugation parameters must be provided [46], leading to a rather simple process once the appropriate conditions are found. The appropriate conditions are based on the presence of large and low-charge counter ions and working at a pH where both AuNPs and proteins are close to electrostatic stability (just below −30 mV for AuNPs and below −10 mV for proteins). Additionally, working close to saturation conditions facilitated cooperative adsorption of proteins, while a larger excess of proteins would have induced a rapid and disordered protein corona formation. In these conditions, the system reaches a final state in which clustering of proteins onto the AuNP surface stabilizes the proteins at the NP surface in a well described phenomenon named hardening of the protein corona, where initially loosely bound proteins evolve to a permanent coating of proteins thanks to coordination bonds with the NP surface and crowding effects between proteins [18].

Viral protein corona formation was monitored first by UV-Vis, DLS, and Zeta Potential. After conjugation, the S-AuNP-VLPs presented a redshift of the SPR peak over 5 nm (Figure 1B), consistent with an almost full coating of the NP surface with proteins. The surface charge of the NPs was also measured: the −40 mV zeta potential for naked (citrate stabilized) AuNP decreased to −18 mV after conjugation with the S protein and purification (Figure 1C). Note that the zeta potential value for free protein dissolved in the same buffer was −12 mV. The hydrodynamic diameter measured by DLS also increased upon conjugation (from 57 to 84 nm), which is in agreement with the UV-Vis spectroscopy measurements and attributed to the formation of a dense monolayer of S protein on top of the AuNP (Figure 1E). No further shifts of the SPR peak post-conjugation were observed after a month (Figure 1F), indicating the high stability of the formed conjugates. Viral proteins conserved their structure during the process—indeed, immobilized proteins tend to be more stable than when free in solution [47].

To further characterize the viral protein corona formed at the NP surface, we carried out aggregation and corrosion tests on the S-AuNP-VLPs. First, we monitored their stability against aggregation in a 200 mM NaCl solution, and second, we assessed Au core corrosion with NaCN. NaCl induces AuNP aggregation due to the screening effects of ions in solution on the NP surface charge, which is responsible for the electrostatic repulsion that keeps NPs apart. However, when NPs are coated by proteins, their surface becomes protected against aggregation thanks to the steric (entropic) repulsion provided by the absorbed proteins [48]. In our case, it was clearly observed that the optical properties of S-AuNP-VLPs were not modified after NaCl addition. However, exposure of non-conjugated AuNPs to NaCl resulted in an increased optical extinction in the region from 600 to 800 nm of the UV-Vis spectra as a consequence of their aggregation (Figure 2A). The tightness or density of the protein coverage was further determined by NP digestion experiments. These consisted of the exposure of the conjugated and naked AuNPs to 15 mM sodium cyanide (NaCN) [49,50]. In the case of citrate-stabilized AuNPs, their surface was readily available for NaCN digestion, resulting in a fast drop in their absorbance. On the contrary, the dense layer of proteins present on S-AuNPs-VLPs protected the gold core, hampering the penetration and induced corrosion of NaCN (Figure 2B). As a control, we also prepared murine serum albumin protein coronas onto 50 AuNPs (MSA-AuNPs-VLPs). These conjugates are a model of NP-hard protein corona when incubated with a 10 times theoretical concentration protein excess [18]. The time needed for naked AuNP complete dissolution was about 25 min, while S and MSA protein coronas considerably slowed this process (Figure 2B). The differences between S and MSA indicate that the S layer is not as dense as the MSA layer, probably due to the relative protein structure and/or the low excess for viral protein conjugation employed. 

The interaction between the AuNP surface and the S protein was also interrogated by molecular dynamics. To gain a more comprehensive understanding of the detailed binding process between spike proteins and AuNPs, Dissipative Particle Dynamics (DPD) was employed. This method simplifies the system using Coarse-Grained (CG) models, which not only reduces computational costs but also extends simulation times. Within this framework, S proteins were represented by three types of CG beads (High = H, Medium = M, and Low = L) based on their interaction with the AuNP (details in Methods). Simulations revealed that S proteins primarily attached to the AuNP surface through their H beads within the first 25 ns. Interestingly, S proteins then tilted away from their initial upright position, suggesting a potential rearrangement, possibly influenced by their interactions with other spike proteins. To further study this phenomenon, the radial distribution function (gofr) between the center of masses (COM) of the AuNP and the S protein was measured. The gofr function essentially depicts the probability of finding a specific bead at a certain distance from the AuNP COM. Figure 3A provides a visual representation of the distinct stages of interaction between S proteins and the AuNP. At the beginning (left), the S proteins are far away from the COM of the AuNP, indicating minimal interaction between them. In contrast, as simulation time progresses (right), the protein becomes adsorbed onto the AuNP surface, indicating a significant interaction between the two entities. Moreover, we can also see that many S proteins are near the COM within their designated region (H bead). Figure 3B offers further insights into the dynamic interaction among S proteins. The snapshots depicted illustrate the evolution of the system at various simulation times, centering attention to one specific spike (highlighted in green color). Once the proteins adhere to the AuNP surface, they undergo a reorganization process. As the simulation progresses, these proteins gradually aggregate, forming clusters on the AuNP surface. This observation suggests that the binding process between S proteins and AuNPs is not static, but rather a delicate balance between the S protein’s affinity for the AuNP and the interactions among neighboring S proteins. 

Finally, the morphology and distribution of prepared conjugates were analyzed through scanning transmission electron microscopy (STEM) and scanning electron microscopy (SEM). As observed in the TEM images, the distribution of citrate-stabilized AuNPs tends to form more and bigger NP assemblies, likely due to stronger electrostatic repulsion with the carbon support film, while S-AuNPs-VLPs are more dispersed in the substrate and less aggregated (Figure 4). In addition, in TEM, where proteins are transparent to the high- energy electron beam, the AuNPs of the VLPs do not touch each other, while for the citrate-stabilized samples, there is direct contact between NPs-NPs. In contrast, in all SEM images, both types of NPs appear in contact since both NP and protein are opaque to the lower-energy electron beam.

### 3.2. Evaluation of SARS-CoV-2-Specific Humoral and Cellular Immune Responses Induced by AuNP-VLPS Presenting the S Protein in BALB/c Mice

Once the formation of a self-assembled viral protein corona was confirmed, its immunogenicity was assessed in BALB/c mice, comparing with similar amounts of free protein and Comirnaty^®^ (Pfizer-BioNTech mRNA COVID-19 vaccine). The SC-Borate solution and MSA-AuNPs were prepared as negative controls. The immunization schedule is shown in Figure 5A. The animals were distributed into five different groups: S-AuNP-VLPs (group A), free S protein (group B), BioNTech vaccine (group C), SC-Borate buffer (group D), and MSA-AuNPs (group E). Except for Comirnaty^®^, the immunogens were applied with InjectTM Alum as an adjuvant. A total of eight mice per group were immunized at days 0 and 15. Two weeks after the second immunization, animals were sacrificed. We assessed if mice immunized with AuNPs-VLPs presenting the S protein could induce SARS-CoV-2 binding and neutralizing antibodies. Induced IgG antibodies in murine sera were measured by ELISA. As noted in Figure 5B, groups A, B, and C exhibited robust production of IgG antibodies. The Comirnaty^®^ immunized group elicited a higher concentration of IgG in comparison with S-AuNP-VLPs and SARS-CoV-2 free protein-immunized mice (*p* < 0.05). Notably, the antibody levels in group B (free protein) were significantly lower than those in AuNP-VLPs and Comirnaty^®^ (*p* < 0.0001) groups. As expected, groups D and E showed undetectable anti-SARS-CoV-2 IgG production. Next, the production of neutralizing antibodies was determined. In concordance with the total IgG anti-S levels, we observed that the titer of neutralizing antibodies followed the same trend. Thus, the neutralizing responses induced by S-AuNP-VLPs and Comirnaty^®^ were comparable [51], and both were superior to those induced by the free protein; the titers of neutralizing antibodies were more than two-fold higher in VLP-immunized animals compared to animals receiving the free protein. Importantly, during the immunization process, none of the vaccinated animals exhibited any observable side effects, such as fever, symptoms of discomfort, or weight loss.

Optimal vaccine-induced immunity varies across infectious diseases, and strategic induction of T cells is often desirable for effective vaccines. In terms of the cellular response, the activation of T cells plays a crucial role in modulating the activation of other immune cells, as well as the response against viral infections and inflammation [52]. T-cell responses have become an important focus in understanding long-term protection from COVID-19. Unlike antibodies, which decline more rapidly and offer less cross-protection against variants, T-cell immunity appears to play a significant role in ongoing protection against severe outcomes such as hospitalization and death [53,54].

Regarding SARS-CoV-2-specific T-cell immune responses, one of the most used methods to evaluate the antigen-specific T-cell responses in mice is to measure the IFN-γ secretion, after stimulation of spleen cells with the SARS-CoV-2 antigen. Thus, the levels of the IFN-γ secreted by splenocytes isolated from treated mouse spleens were determined by ELISA assay, after a 72-h stimulation with the full sequence of the S recombinant protein of the SARS-CoV-2 antigen. In mice, spleen cell cultures typically consist of around 100 million splenocytes, which are mononuclear white blood cells located in or derived from the spleen. T cells constitute approximately 25% of the total splenocyte population, providing a reliable readout of antigen-specific T cells secreting IFN-γ in response to antigen restimulation [55,56]. As shown in Figure 5D, a high level of IFN-γ was observed in the S-AuNPs-VLPs immunized group. On the contrary, very low levels (or undetectable secretion) of IFN-γ were observed in the rest of the immunized mouse groups, including the Comirnaty^®^ vaccinated group.

## 4. Discussion

In this study, we developed deactivated virus-like particles comprising gold cores enveloped by coronas of the full SARS-CoV-2 spike protein. Using BALB/c mice as a model, we assessed the immunogenicity of these S-AuNPs-VLPs. Our findings reveal that S-AuNPs-VLPs consistently boost antigen-specific antibody responses compared to the free S protein in solution. This augmentation includes not only higher IgG titers and enhanced neutralizing capacity of antibodies but also activation of T-cell responses. Compared to mRNA COVID-19 vaccines, S-AuNPs-VLPs induced comparable binding and neutralizing antibody titers but superior T-cell responses. S-AuNPs-VLPs were designed using a novel conjugation strategy previously described by our research group [30]. In this approach, S proteins were attached to the AuNP surface, forming ordered domains while fully retaining their antigenic properties capable of triggering a robust immune response. The results presented here demonstrate that S-AuNPs-VLPs induce a strong immune response, generating high levels of antibodies as well as INF-γ. We optimized the conjugation conditions and ensured colloidal stability to achieve S protein coronas driven by cooperative adsorption, leading to their organized domains on the AuNP surface. The fine-tuning of protein functionalization achieved in AuNPs to promote a cooperative adsorption of proteins, reaching highly dense protein patterns, would not have been possible with other types of NPs. This is crucial to explain the behavior of our system. In the context of AuNP conjugation, cooperative adsorption is the phenomenon where the protein adsorption onto the surface of the NPs is influenced positively by the presence of previously adsorbed molecules. Cooperative adsorption implies that a protein adsorbed to a surface binds more strongly to it if it is surrounded by similar proteins thanks to the possibility of building compact arrangements with identical building blocks, increasing coating stability (cooperative effect) as the protein domain grows. This phenomenon favors clustering of proteins versus random positioning. We confirmed that the experimental conditions for synthesizing VLPs preserved the biofunctional properties and stability of the S protein, emphasizing the potential of this approach in vaccine development and the possibility of storage conditions between 2 °C and 8 °C. The AuNP surface–protein interaction must be achieved in an adequate conjugation condition to prevent NP aggregation or protein denaturation. Changes in pH, ionic strength and/or NP and protein concentrations can lead to aggregation of these components in solution. As known, when proteins lose their native structure, they unfold and expose hydrophobic regions that are usually buried within the protein. Therefore, once the proteins are denatured, they tend to aggregate with other proteins presents in the solution, aggregating the AuNPs with them [47,57]. This aggregation can lead to the proteins forming larger complexes and linking with other AuNPs. Importantly, the characteristic width of the measured peaks after conjugation remained similar to the one before conjugation, further confirming the formation of homogeneous viral protein coronas. Otherwise, in the event of a non-homogeneous conjugation process, the resulting conjugates would display varying degrees or types of antibody coating, and consequently, UV-Vis, DLS, and Z-potential peaks would exhibit broader profiles compared to those for non-conjugated NPs. The observed peaks, narrow like the initial ones but shifted, indicate the narrow dispersion of both the NP core and the NP coating.

Although current vaccines against coronavirus infection utilize various approaches such as subunit vaccines, viral vectors, DNA, and mRNA sequences targeting the viral S protein or other viral epitopes [58,59,60], there is a clear need to continue exploring new tools to design nano-vaccines that increase the longevity of the immune response, one of the main remaining challenges [61]. In general, AuNPs are considered biocompatible and have low toxicity at appropriate doses. Extensive studies have shown that AuNPs are well-tolerated in vivo, making them a reliable choice for vaccine platforms [10,62,63]. Compared to other VLP platforms produced in eukaryotic and prokaryotic cell lines to entrap therapeutic cargo, using AuNPs as the core reduces major drawbacks such as high production costs, difficulty in scaling up, time consumption, and low yield. In the specific case of SARS-CoV-2-AuNPs-VLPs, diverse studies have demonstrated high immunogenicity, producing long-lasting antigen-specific IgG response [29,64]. Moreover, they exhibit thermal stability even with temperature fluctuations, which reduces costs by eliminating the need for a cold chain [64].

In our study, we emphasized the importance of selecting the appropriate size of AuNPs and using the full sequence of the S protein to ensure the production of high levels of both antibodies: binding and neutralizing. The presence of binding antibodies indicates the ability to recognize and attach to the protein antigen coated onto the AuNP surface, while the levels of neutralizing antibodies demonstrate the capacity to block viral replication within an in vivo model. Note that some studies reveal an extended presence of antibodies in human serum against the spike protein component in comparison to other viral components such as nucleocapsid or nucleocapsid RNA binding domain (N-RBD) [65,66] after infection, another reason to select the S protein as vaccine training element.

For an effective response after vaccination, it is essential to balance the activation of both the humoral and cellular immune responses. Indeed, an optimal T-cell response can enhance the overall efficacy of the vaccine by providing broader and more durable immunity in COVID disease [67]. T cells exhibit the ability to differentiate into helper lymphocytes (CD4+) and cytotoxic cells (CD8+), contributing substantially to immune defense mechanisms. Cytotoxic CD8+ T cells can recognize and kill virally infected cells, thus providing local control of viral infection in tissues. CD4+ T cells provide help, especially to B cells, required for the development of a mature antibody response. Upon differentiation, T cells can secrete various cytokines to modulate the behavior of other immune cells and amplify the immune response. Among the diverse repertoire of cytokines, interferon-gamma (IFN-γ) has a key role, contributing to protection from viral and bacterial infections and regulating effector cells in both innate and adaptive immunity [68]. This observed improvement in VLPs immunization potency can be attributed to several factors. Firstly, antigen delivery to the lymph nodes, where a high number of antigen-presenting cells reside, is favored when using a nanoparticle (<100 nm) as a vehicle [20,21]. Additionally, the particulate nature of the VLPs may have facilitated improved cellular uptake and enhanced complement activation [10,69]. Moreover, the VLPs may have been efficiently presented by follicular dendritic cells, further enhancing their immunogenicity [62] and value for further exploration.

The findings of this study will significantly support the development of vaccines not only against SARS-CoV-2 infection but also against other severe respiratory viruses. In our case, the insights into optimal antigen selection (S protein), adjuvant use, and AuNP size can be applied to enhance the immunogenicity and durability of vaccines for other respiratory pathogens. Last but not least, while the use of AuNP virus-like particles shows promising results in new vaccine development technology, and despite AuNPs’ high biocompatibility, there are some limitations related to their accumulation, biodegradation and excretion (via the hepatobiliary route), and clearance by the mononuclear phagocytosis system that need to be determined before parenteral administration of AuNPs to humans.

## 5. Conclusions

Various vaccination strategies for SARS-CoV-2, including mRNA technology, protein subunits, and VLPs, have been employed to date. However, the emergence of new variants has highlighted the necessity of developing new tools to address the challenges posed by increased transmissibility, virulence, or immune evasion associated with these variants. In this context, VLPs based on gold cores are a promising tool for rapidly generating a new vaccine providing a robust immune response. In general, VLPs offer a range of significant benefits, with one of the most important being their ability to stimulate both cellular and humoral immune responses. The superior performance of S-AuNPs-VLPs in inducing T cell-mediated immunity compared to mRNA is of interest. T-cell responses can augment the protective capacity, in particular, to prevent severe disease. Furthermore, enhanced T-cell responses may contribute to more durable antibody responses. This is important, as antibody responses induced by current COVID vaccines decay notoriously rapidly. Importantly, the adaptable nature of VLPs, particularly when utilizing AuNPs, enables rapid targeting of new strains through modified sequences of the S protein or other target sequences to be adsorbed on the gold surface in a very simple manner. Last but not least, the extra stability provided by the absorption of protein onto inorganic surfaces facilitates storage under standard conditions. With new next-generation vaccines being continuously developed, biologists and immunologists must engage with chemists, materials scientists, and engineers in the collaborative, coordinated development of universal methods to engineer potent next-generation vaccines.

## Figures and Tables

**Figure 1 vaccines-12-00829-f001:**
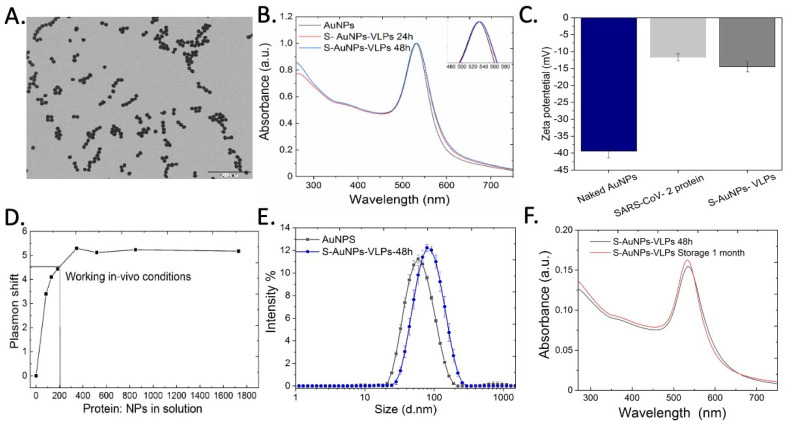
**Characterization of the S-AuNPs-VLPs**. (**A**) Representative TEM image of the ~50 nm AuNPs. (**B**) Time-dependent UV-Vis spectra of the S-AuNPs-VLPs compared to citrate-stabilized AuNPs. (**C**) (Z-potential measurements of citrate-stabilized AuNPs, S protein, and S-AuNPs-VLPs (n = 3, mean with SD). (**D**) Langmuir-type isothermal adsorption curves. (**E**) Dynamic light scattering (DLS) characterization of S-AuNPs-VLPs compared to citrate-stabilized nanoparticles. (**F**) UV-Vis spectra of the S-AuNPs-VLPs after 1 month of conjugation.

**Figure 2 vaccines-12-00829-f002:**
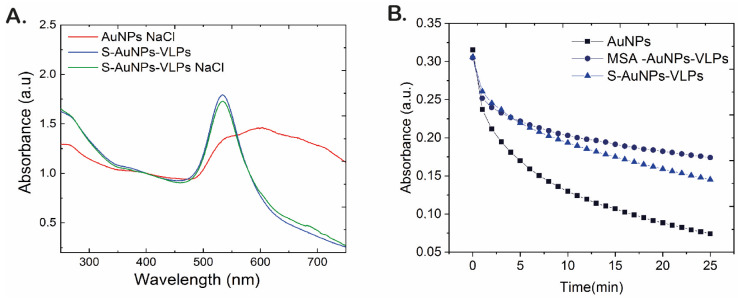
**AuNPs and S-AuNPs-VLPs stability**. (**A**). UV-vis spectra of naked and conjugated AuNPs treated with 200 mM NaCl. (**B**). Time evolution degradation of AuNPs incubated with 15 mM of NaCN.

**Figure 3 vaccines-12-00829-f003:**
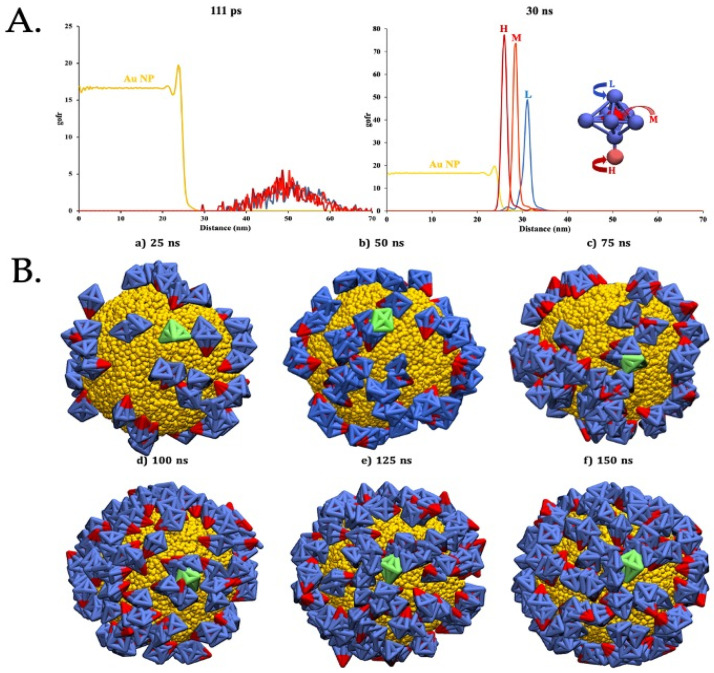
**Molecular simulation of AuNPs and S proteins** (**A**). gofr for the AuNP (in dark yellow) and the H (dark red), M (red), and L (blue) from the S proteins at short times (**left**) and longer times (**right**). (**B**). Time-dependent evolution of S protein aggregation captured at (**a**) 25 ns, (**b**) 50 ns, (**c**) 75 ns, (**d**) 100 ns, (**e**) 125 ns, and (**f**) 150 ns. The S protein depicted in green serves as the focal point, providing insights into the dynamic behavior of S protein over time. Water beads are removed for the sake of clarity.

**Figure 4 vaccines-12-00829-f004:**
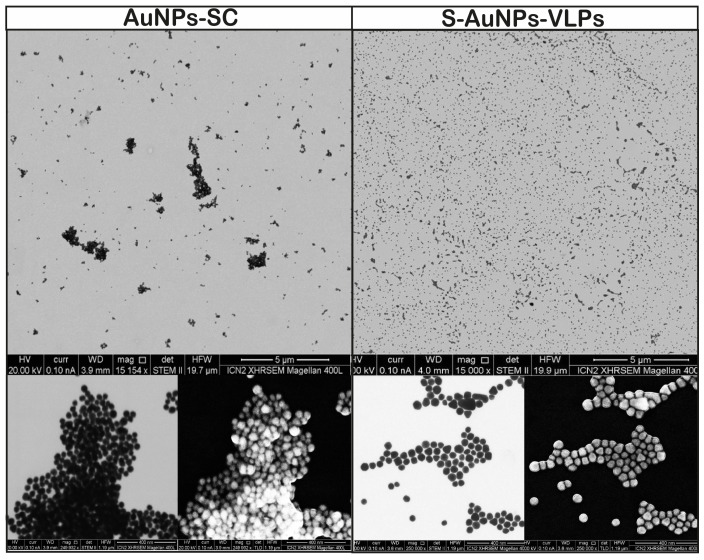
**Microscopy visualization of AuNPs and surface-modified AuNPs-VLPs** Scanning transmission electron microscopy (STEM) and scanning electron microscopy (SEM) images of gold nanoparticles and S-AuNPs-VLPs produced by cooperative adsorption.

**Figure 5 vaccines-12-00829-f005:**
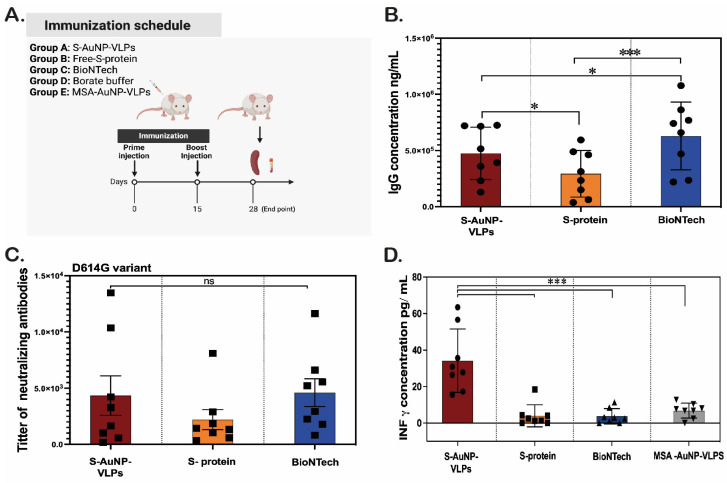
**Induction of SARS-CoV-2-specific antibodies and T-cell responses after AuNPs-VLPs immunization in BALB/c mice.** (**A**)**.** Immunization schedule. Eight mice (male *n* = 4 and female *n* = 4) in each group were immunized intramuscularly (i.m) twice with 10 ug of attached S protein on AuNPs (S-AuNPs-VLPs), 10 ug of free protein, or 100 ng mRNA of Comirnaty^®^ vaccine. One group of mice was immunized with Borate buffer (naïve group, no VLP, negative control) and another group with murine albumin VLPs (MSA, unrelated VLP). The homologous prime-boost interval was 2 weeks. The endpoint of this trial was on day 28. Sera and spleens were collected for ELISA assays. Sera were also collected for neutralization assays. (**B**). SARS-CoV-2 spike-specific antibodies induced by S-AuNPs-VPLs, free S protein, and BioNTech. ELISA assay was performed to analyze anti-SARS-CoV-2 spike IgG. (**C**). Neutralizing antibody titer against Wuhan strain. (**D**). SARS-CoV-2 spike-specific T-cell immune responses induced by AuNPs-VLPs. ELISA assay was performed to measure the IFN-γ secreting splenocytes after stimulation with free S protein. No detectable levels of IgG and neutralizing antibodies were determined in either negative control (Borate buffer or MSA-AuNPs-VLPs). Each dot shows the data from an individual animal. * *p* < 0.05 or *** *p* < 0.0001, *ns*: no significative differences. Tukey’s multiple comparisons test followed by two-way ANOVA.

**Table 1 vaccines-12-00829-t001:** Interacting values (SRP) for water (W), AuNP (Au), and S protein (L, M, H).

SRP	W	Au	S Protein
W	25	120	40
Au		1	55 (L) 35 (M) 15 (H)
SP			55

## Data Availability

Data is contained within the article or Appendix A.

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
