# Peer review of "Gold Nanoparticle Virus-like Particles Presenting SARS-CoV-2 Spike Protein: Synthesis, Biophysical Properties and Immunogenicity in BALB/c Mice"

_vaccines, 2024, doi:10.3390/vaccines12080829_

Round 1

Reviewer 1 Report

Comments and Suggestions for Authors

This study reported the work with gold nanoparticles assembled with SARS-CoV-2 spike protein and engineered into virus-like particles, showing enhanced immunogenicity in BALB/c mice compared to free S protein in solution. These S-AuNPs-VLPs induced robust antibody and particularly, T-cell responses, suggesting their potential as effective antigen-delivery systems.

This article would be of value to readers with these suggestions taken for improvement.

Line 272, elaborate on how protection of virus growth is measured.

Line 273, why the production of IFN-gamma is only analyzed in spleen? What about the circulating IFN-gamma?

Line 299, Is the separation between spike proteins in SARS-CoV-2 on naturally occurring SARS-CoV-2 is around 25 nm too? Then how the virus effectively activate immune response?

Provide data or discussion on the superiority of S-AuNPs-VLPs compared to other types of VLPs.

Comments on the Quality of English Language

Overall high quality

Author Response

We sincerely appreciate the editors' consideration of our work for publication. We have revised the manuscript to address referees' concerns. We believe this updated version highlights the significance of our research. Thank you once again for considering our work, and we hope this revised version meets your expectations.

Reviewer 1

This study reported the work with gold nanoparticles assembled with SARS-CoV-2 spike protein and engineered into virus-like particles, showing enhanced immunogenicity in BALB/c mice compared to free S protein in solution. These S-AuNPs-VLPs induced robust antibody and particularly, T-cell responses, suggesting their potential as effective antigen-delivery systems.

This article would be of value to readers with these suggestions taken for improvement.

Line 272, elaborate on how protection of virus growth is measured.

We thank the reviewer for their thoughtful and thorough review. To better describe and clarify our results, we have made some changes to the main text to enhance its value for the readers and added some references.

In this study, we have evaluated in vivo the specific SARS-CoV2 immunogenicity after BALB/c mice immunization with S-AuNPs-VLPs.  Before to evaluate protection in vivo in mice we have to know if our vaccine candidate induces specific SARS-CoV-2 immune responses.  Besides that, we have measured in vitro virus neutralization, which is extremely well correlated with in vivo protection in humans (Khoury et al. 2021).

Khoury, D. S. et al. Neutralizing antibody levels are highly predictive of immune protection from symptomatic SARS-CoV-2 infection. Nat. Med. 2021 277 27, 1205–1211 (2021)

This comment with its reference has been introduced in the last paragraph of the  introduction section.

Line 273, why the production of IFN-gamma is only analyzed in spleen? What about the circulating IFN-gamma?

Thanks for the revision, to explain the differences between  the INF-gamma quantification in spleen and blood, we introduced some changes in the main text. One of the most used method to evaluate the Antigen-specific T-cell responses in mice, is to measure the IF- É£ secretion, after spleen cells stimulation with the SARS-CoV-2 antigen.

In our case with full length SARS-CoV-2 Spike recombinant protein. In a mouse, the spleen cell culture is composed of around 100 million splenocytes. Splenocytes are mononuclear white blood cells (WBCs) derived from or situated in the spleen. T cells typically comprise roughly 25% of the total splenocyte population. We remove the spleen tissue and homogenize, count the total number of cells and stimulate them with the specific antigen (See material and methods section).

Regarding the circulating IFN-É£ we do not know from where it does comes from (i.e. are they from ag-specific cells?). As we pointed out above, the spleen contains large amounts of T cells and we specifically stimulate the Ag-specific ones thereby creating a reliable read-out of Ag-specific T cells secreting IFN-É£ in response to Ag-restimulation (Wang et al. 2023) (Huang, Ma, and Wu 2006).

Wang, X. et al. Vaccine-induced protection against SARS-CoV-2 requires IFN-γ-driven cellular immune response. Nat. Commun. 14, 1–15 (2023).

Huang, J., Ma, R. & Wu, C. you. Immunization with SARS-CoV S DNA vaccine generates memory CD4+ and CD8+ T cell immune responses. Vaccine 24, 4905–4913 (2006).

This comment with its reference has been introduced in the last paragraph before the discussion.  

Regarding SARS-CoV-2-specific T-cell immune responses, one of the most used method to evaluate the Antigen-specific T-cell responses in mice, is to measure the IFN- É£ secretion, after spleen cells stimulation with the SARS-CoV-2 antigen. Thus, the levels of the IFN-γ secreted by splenocytes isolated from treated mice spleens were determined by ELISA assay, after a 72-hour stimulation with the full sequence of the S recombinant protein of the SARS-CoV-2 antigen. In mice, spleen cell cultures typically consist of around 100 million splenocytes, which are mononuclear white blood cells located in or derived from the spleen. T cells constitute approximately 25% of the total splenocyte population, providing a reliable readout of antigen-specific T cells secreting IFN-γ in response to antigen restimulation (Wang et al. 2023) (Huang, Ma, and Wu 2006).

Line 299, Is the separation between spike proteins in SARS-CoV-2 on naturally occurring SARS-CoV-2 is around 25 nm too? Then how the virus effectively activate immune response?

As stated in the manuscript, differently to the vast majority of RNA viruses, SARS-CoV-2 present a large separation between spike proteins units (around 25 nm), which might explain the observed inefficient and short-lived neutralizing antibody response after infection with SARS-CoV-2.

This has been added to the text for clarification, now the text reads as:

Before:

Remarkably, it presents a particular challenge; it has been reported that differently to the vast majority of RNA viruses, the separation between spike proteins in SARS-CoV-2 is around 25 nm, far from the 5-10 nm needed for optimal immune activation, which might explain the inefficient and short-lived neutralizing antibody responses, after infection with SARS-CoV-2 (Bachmann et al. 2021). Under these circumstances, presenting a dense mosaic of S protein to train our immune system is particularly appealing. The AuNP size was chosen to mimic SARS-CoV-2 size and to accommodate a substantial number of proteins on their Surface.

After:

Remarkably, it presents a particular challenge; it has been reported that differently to the vast majority of RNA viruses, the separation between spike proteins in SARS-CoV-2 is around 25 nm, far from the 5-10 nm needed for enhanced immune activation, since epitope repetition boost immunogenicity (Bachmann et al. 1993) (Tariq et al. 2022) (Ogrina et al. 2023), which might explain the inefficient and short-lived neutralizing antibody responses, after vaccination and infection with SARS-CoV-2 (Bachmann et al. 2021), highlighting the importance of structure in antigen presentation. Under these circumstances, presenting a dense mosaic of S protein to train our immune system is particularly appealing. The AuNP size was chosen to mimic SARS-CoV-2 size and to accommodate a substantial number of proteins on their surface.

Some references have been updated:

Bachmann, M. F., Mohsen, M. O., Zha, L., Vogel, M. & Speiser, D. E. SARS-CoV-2 structural features may explain limited neutralizing-antibody responses. npj Vaccines vol. 6 (2021).

Bachmann, M. F. et al. The influence of antigen organization on B cell responsiveness. Science (80-. ). 262, 1448–1451 (1993).

Tariq, H., Batool, S., Asif, S., Ali, M. & Abbasi, B. H. Virus-Like Particles: Revolutionary Platforms for Developing Vaccines Against Emerging Infectious Diseases. Frontiers in Microbiology vol. 12 (2022).

Ogrina, A. et al. Bacterial expression systems based on Tymovirus-like particles for the presentation of vaccine antigens. Front. Microbiol. 14, (2023).

Provide data or discussion on the superiority of S-AuNPs-VLPs compared to other types of VLPs.

Thank you to the reviewers for their comments and suggestions. We have revised the discussion section to enhance clarity,  understanding and updating some references.

The fine tuning of protein functionalization achieved in AuNPs to promote a cooperative adsorption of proteins, reaching highly dense protein patterns would not have been possible with other types of nanoparticles. This is crucial to explain the behavior of our system. In general, AuNPs are considered biocompatible and have low toxicity at appropriate doses. Extensive studies have shown that AuNPs are well-tolerated in vivo, making them a reliable choice for vaccine platforms (Dykman 2020) (Sengupta et al. 2022) (Carabineiro 2017).

The high physicochemical stability and resistant to degradation under both storage conditions and within biological environments ensures that the AuNPs-VLPs remain effective over time without the need of complex storage conditions. 

Compared to other VLPs platforms produced in eukaryotic and prokaryotic cell lines to entrap therapeutic cargo, using gold nanoparticles as the core reduces major drawbacks such as high production costs, difficulty in scaling up, time consumption, and low yield. In the specific case of  SARS-Cov-2- AuNPs- VLPs diverse studies have demonstrated high  immunogenicity producing  long-lasting antigen-specific IgG response (Sekimukai et al. 2020) (Farfán-Castro et al. 2024). Moreover, they exhibit thermal stability, even with temperature fluctuations, which reduces costs by eliminating the need for a cold chain (Farfán-Castro et al. 2024).

The references have been actualized in the main text:

Dykman, L. A. Gold nanoparticles for preparation of antibodies and vaccines against infectious diseases. Expert Rev. Vaccines 00, (2020).

Sengupta, A., Azharuddin, M., Al-Otaibi, N. & Hinkula, J. Efficacy and Immune Response Elicited by Gold Nanoparticle-Based Nanovaccines against Infectious Diseases. Vaccines 10, 1–22 (2022).

Carabineiro, S. A. C. Applications of gold nanoparticles in nanomedicine: Recent advances in vaccines. Molecules vol. 22 (2017).

Sekimukai, H. et al. Gold nanoparticle-adjuvanted S protein induces a strong antigen-specific IgGesponse against severe acute respiratory syndrome-related coronavirus infection, but fails to induce protective antibodies and limit eosinophilic infiltration in lungs. Microbiol. Immunol. 64, 33–51 (2020).

Farfán-Castro, S. et al. Synthesis and evaluation of gold nanoparticles conjugated with five antigenic peptides derived from the spike protein of SARS-CoV-2 for vaccine development. Front. Nanotechnol. 6, 1–15 (2024).

Reviewer 2 Report

Comments and Suggestions for Authors

The manuscript of the article vaccines-3083062 «Gold nanoparticles virus-like particles presenting SARS-CoV2 spike protein: Synthesis, biophysical properties and immunogenicity in BALB/c mice» introduces a very interesting approach to the vaccine production based on gold nanoparticles (NPs), which makes it possible not only to get the possibility of the rapid vaccines production that retain their properties under fairly simple storage and transportation conditions, which is very relevant from the point of view of their use. But, what is fundamentally more important, these vaccines are capable of activating both the humoral and cellular immune responses. Undoubtedly, this approach will significantly support the development of vaccines against various virulent respiratory viruses.

Nevertheless, this work deserves attention, but in present form the manuscript cannot be recommended for publication, it requires a major revision.

-       «The interaction between proteins and inorganic surfaces, particularly gold nanoparticles, has been extensively investigated under the term Protein Corona [13]» need to add more up-to-date references

-       Taking into account the variety of processes in the body mediated by the immune system, I would like to advise the authors to take a more rigorous look at the use of the terms "immunological outcome and immune response", as well as "immunologic response" and prescribe which of the processes they mean in each of these cases.

-       It would also be good to describe in more detail what is meant by "cooperative absorption" in this context. Cooperative effects can affect both the binding affinity and the manifestation of the bound molecule properties.

-       Unfortunate name «AuNP functionalization with…».

-       The process of obtaining S-AuNPs-VLPs should be described in more detail. In the process of conjugation, whether the covalent bond of the Au-core and the SARS-CoV-2 spike protein occur. Since the ratio of 1:200 is assumed in the production of S-AuNPs-VLPs, if without a covalent bond, to what which the authors attribute the stability of this system under conditions of low concentrations and in the presence of competitive compounds in vivo.

-       The abbreviation should be deciphered at the first mention (SRP, DPD…).

-       English: “protein dispersed in a 10 mM borate-conjugating medium solution”

-       There is a discrepancy between the figures and the caption to the figures, as well as the mention of the figures in the text (fig.1).

-       A single terminology should be adhered to (Murine albumin, mouse albumin)

-       The statement «Note that if proteins were denatured as a consequence of the interaction with the AuNP surface, they would inevitably lead to aggregation and detectable NP cross-linking.» seems unfounded.

Comments on the Quality of English Language

the inconsistent expressions (protein dispersed), difficult to understand due to indiscriminate use of terms

Author Response

We sincerely appreciate the editors' consideration of our work for publication. We have revised the manuscript to address referees' concerns. We believe this updated version highlights the significance of our research. Thank you once again for considering our work, and we hope this revised version meets your expectations.

 -       «The interaction between proteins and inorganic surfaces, particularly gold nanoparticles, has been extensively investigated under the term Protein Corona [13]» need to add more up-to-date references

The following references have been added to support the investigation in the field of protein corona formation.

 Liu, J.; Peng, Q. Protein-gold nanoparticle interactions and their possible impact on biomedical applications. Acta Biomater. 2017, 55, 13–27, doi:10.1016/j.actbio.2017.03.055.

García-Álvarez, R.; Hadjidemetriou, M.; Sánchez-Iglesias, A.; Liz-Marzán, L.M.; Kostarelos, K. In vivo formation of protein corona on gold nanoparticles. the effect of their size and shape. Nanoscale 2018, 10, 1256–1264, doi:10.1039/c7nr08322j.

Nandakumar, A.; Wei, W.; Siddiqui, G.; Tang, H.; Li, Y.; Kakinen, A.; Wan, X.; Koppel, K.; Lin, S.; Davis, T.P.; et al. Dynamic Protein Corona of Gold Nanoparticles with an Evolving Morphology. ACS Appl. Mater. Interfaces 2021, 13, 58238–58251, doi:10.1021/acsami.1c19824.

Dridi, N.; Jin, Z.; Perng, W.; Mattoussi, H. Probing Protein Corona Formation around Gold Nanoparticles: Effects of Surface Coating. ACS Nano 2024, 18, 8649–8662, doi:10.1021/acsnano.3c08005.

-       Taking into account the variety of processes in the body mediated by the immune system, I would like to advise the authors to take a more rigorous look at the use of the terms "immunological outcome and immune response", as well as "immunologic response" and prescribe which of the processes they mean in each of these cases.

We agree and we have taken a more rigorous look at the use of these terms in the revised version. We have modified the manuscript to clarify the immune response triggered by using the VLPs.

-       It would also be good to describe in more detail what is meant by "cooperative absorption" in this context. Cooperative effects can affect both the binding affinity and the manifestation of the bound molecule properties.

The manuscript has been updated to clarify this aspect. The following paragraph has been added to the discussion section.

In the context of gold nanoparticles conjugation, the cooperative adsorption is the phenomenon where the protein adsorption onto the surface of the nanoparticles is influenced positively by the presence of previously adsorbed molecules. Cooperative adsorption implies that a protein adsorbed to a surface bind more strongly to it if it is surrounded by similar proteins thanks to the possibility of building compact arrangements with identical building blocks, increasing coating stability (cooperative effect) as the protein domain grows. This This phenomenon favors clustering of proteins versus random position.

-       Unfortunate name «AuNP functionalization with…».

The subtitle mistakes have been corrected through the text.  

 -       The process of obtaining S-AuNPs-VLPs should be described in more detail. In the process of conjugation, whether the covalent bond of the Au-core and the SARS-CoV-2 spike protein occur. Since the ratio of 1:200 is assumed in the production of S-AuNPs-VLPs, if without a covalent bond, to what which the authors attribute the stability of this system under conditions of low concentrations and in the presence of competitive compounds in vivo.

Adsorption of S protein onto the gold surface is a non-static process in which the system evolves towards an equilibrium, which is influenced at any moment by both the availability of gold surface and the interactions with neighboring S proteins.

Given the appropriate conditions described in the manuscript, the system reaches a final state in which clustering of proteins onto the gold surface stabilizes the proteins at the NP surface in a well described phenomena named hardening of the protein corona where initially loosely bound proteins evolve to a permanent coating of proteins thanks to coordination bonds to the NP surface and crowding effects between proteins (Casals et al. 2010).

This sentence and the appropriate reference has been added to the main text

Note that according to Langmuir-type isothermal adsorption curves (Figure 1D), ratio 1:200 was chosen to ensure high density of proteins needed to guarantee steric hindrance, preventing aggregation and non-specific interactions, whilst cooperative effects are promoted. Stability test on high ionic strength media (Fig 2A) demonstrates that the S protein layer was stable: if desorption has occurred, gold nanoparticles would have aggregated. All in all, the combination of high-affinity binding, multivalent interactions, and cluster-protein distribution contributes to the robustness of the S-AuNPs-VLPs system.

Reference

Casals, E., Pfaller, T., Duschl, A., Oostingh, G. J. & Puntes, V. Time evolution of the nanoparticle protein corona. ACS Nano 4, 3623–3632 (2010).

-       The abbreviation should be deciphered at the first mention (SRP, DPD…).

The abbreviation mentioned above were corrected in the main text for major clarity.

Done

-       English: “protein dispersed in a 10 mM borate-conjugating medium solution”

The phrase was corrected in the main text for: These AuNPs were subsequently added dropwise to the lyophilized viral protein dissolved in borate buffer 10mM - for 48 hours under gentle stirring at 4 °C.

-       There is a discrepancy between the figures and the caption to the figures, as well as the mention of the figures in the text (fig.1).

 Corrected.

-       A single terminology should be adhered to (Murine albumin, mouse albumin)

Corrected.

-       The statement «Note that if proteins were denatured as a consequence of the interaction with the AuNP surface, they would inevitably lead to aggregation and detectable NP cross-linking.» seems unfounded.

The interaction gold nanoparticle surface-protein must be done in an adequate conjugation condition to prevent nanoparticles aggregation or protein denaturation. Changes in pH, ionic strength and/or NPs- protein concentrations can lead to aggregation of these components in solution. As known, when proteins lose their native structure, they unfold and expose hydrophobic regions that are usually buried within the protein. Therefore, once proteins are denatured, they tend to aggregate with other proteins presents in the solution aggregating with them the AuNPs. This aggregation can lead to the proteins forming larger complexes and linking with other AuNPs.  The text has been modified to clarify this point and two references have been added.

Reviewer 3 Report

Comments and Suggestions for Authors

This paper describes the synthesis, biophysical and immunogencity characterization of gold NP VLP that presenting SARS-CoV-2 S-protein. The paper is well written in general, and it may be considered for publication after addressing the following questions. 

1) For Figure 8A, what does 10 ug mean for S-AuNP-VLPs, Free -S- 476 protein, or Comirnaty, respectively?10 ug of protein or nucleic acid or 10 ug total particles? It is not very clear here.

2) Any potential side effect of this gold NP VLP based vaccine?

Author Response

We sincerely appreciate the editors' consideration of our work for publication. We have revised the manuscript to address referees' concerns. We believe this updated version highlights the significance of our research. Thank you once again for considering our work, and we hope this revised version meets your expectations.

1. For Figure 8A, what does 10 ug mean for S-AuNP-VLPs, Free -S- protein, or Comirnaty, respectively?10 ug of protein or nucleic acid or 10 ug total particles? It is not very clear here.

        The figure caption was rewritten to clarify the quantities of each immunogen.

2. Any potential side effect of this gold NP VLP based vaccine?

The following paragraph has been added to the conclusions section:

Last but not least, while the use of AuNPs virus-like particles shows promising results in new vaccine development technology, and despite AuNPs high biocompatibility, there are some limitations related to their accumulation, biodegradation and excretion (via de hepatobiliary route), and their clearance by the mononuclear phagocytosis system that needs to be determined before parenteral administration of AuNPs to humans.

Round 2

Reviewer 2 Report

Comments and Suggestions for Authors

The main provisions are corrected , which improved the quality of the manuscript. However, numerous editing flaws remain.

Author Response

Dear Editors and Reviewers,

Thank you for the extensive evaluation and comments on our manuscript, ID: vaccines-3083062, titled "Gold nanoparticle virus-like particles presenting SARS-CoV-2 spike protein: Synthesis, biophysical properties, and immunogenicity in BALB/c mice." According to your kind revisions, we have corrected the minor issues in the main text.

We highlight the corrections in the current manuscript and detail each change:

  • Line 23: Corrected "BALC/c" to "BALB/c"
  • Line 27: Corrected "covid" to "COVID-19"
  • Line 86: Corrected “Naïve” to “naïve”
  • Line 107: Corrected “in vivo” to “in-vivo” “SARS-CoV2." to “SARS-CoV-2."
  • Line 108: Corrected “in vitro” to “in-vitro”
  • Line 109: Corrected “in vivo” to “in-vivo”
  • Line 111: Corrected “in vivo” to “in-vivo”
  • Line 121: Changed “protein” to “proteins”
  • Line 132: Corrected “AuNP synthesis” to “Synthesis of AuNPs”
  • Line 146: Added hyphens
  • Line 149: Changed AuNP to AuNPs
  • Line 159: Changed “using” to “with”
  • Line 161: Changed “gold nanoparticles” to “AuNPs”
  • Line 185: Changed “spectrum” to “spectra”
  • Line 186: Removed of
  • Line 193: Bold removed in the word Figure S1
  • Line 195: Bold added to the word Soft Repulsive Parameters
  • Line 233: Changed TEM to STEM
  • Line 234: Added Scanning and STEM
  • Line 235: Changed “gold nanoparticles” to “AuNPs”
  • Line 241: Changed “nanoparticles” to “NPs”
  • Line 242: Changed “nanoparticles” to “NPs”
  • Line 276: Corrected “in vivo” to “in-vivo”
  • Line 306: A comma was removed
  • Line 335: Removed the word Surface plasmon resonance
  • Line 345: Added hyphens
  • Line 390: Changed shifts to shift
  • Line 461: Added the words scanning transmission electron microscopy (STEM)
  • Line 468 and 471: Removed the letter S in the bean word
  • Line 466: Corrected “high energy electron beams” to “high-energy electron beams”
  • Line 471-472: Added Scanning and STEM
  • Line 472: Removed the word naked
  • Line 481: Removed "the" before "MSA-AuNPs"
  • Line 483: Removed hyphens
  • Line 484: Changed “day 0 and day 15” to “days 0 and 15”
  • Line 487: Modified “exhibited a robust” to “exhibited robust”
  • Line 489: Added hyphens
  • Line 491: Added the word “groups”
  • Line 505: Changed gold to AuNPs
  • Line 509: Changed "Sera was collected also" to "Sera was also collected."
  • Line 526: Changed "one of the most used method" to "one of the most used methods."
  • Line 527: Corrected "IFN-É£" to "IFN-γ."
  • Line 532: Added hyphens
  • Lines 535-536: Corrected "very low level (or undetectable) secretion" to "very low levels (or undetectable secretion)."
  • Line 540: Removed (VLPs)
  • Line 545: Changed “vaccine” to “vaccines”
  • Line 556: Changed “nanoparticles” to “NPs”
  • Line 557: ChangedGold nanoparticles” to “AuNPs”
  • Line 558: Changed “nanoparticles” to “NPs”
  • Line 566: Removed extra space in "°C"
  • Line 566: ChangedGold nanoparticle” to “AuNP”
  • Line 567: Changed “nanoparticles” to “NPs”
  • Line 587: Corrected “in vivo” to “in-vivo”
  • Line 589: ChangedGold nanoparticles” to “AuNPs”
  • Line 601: Corrected instances of "in-vivo" to "in vivo".
  • Line 606: Corrected instances of " T-cell " to " T cell ".

Thank you for your attention to these revisions. We believe these changes will significantly improve the clarity and quality of our manuscript. We hope the revised manuscript meets your expectations.
